# Progress of Diabetic Macular Edema after Loading Injection of Anti-Vascular Endothelial Growth Factor Agents in Real-World Cases

**DOI:** 10.3390/medicina58101318

**Published:** 2022-09-21

**Authors:** Hiroko Enomoto, Masahiko Sugimoto, Shin Asami, Mineo Kondo

**Affiliations:** Department of Ophthalmology, Mie University Graduate School of Medicine, Edobashi, Tsu 514-8507, Japan

**Keywords:** anti-vascular epithelial growth factor, diabetic macular edema, loading injection, pro re nata

## Abstract

*Background and Objectives*: To evaluate the recurrence of diabetic macular edema (DME) after loading an injection of anti-VEGF agents by a pro re nata (PRN) protocol using central retinal thickness (CRT) as a re-injection criterion. *Materials and Methods*: This is a retrospective, observational single-center study. DME patients with a central retinal thickness (CRT) over 350 μm received a PRN injection of anti-VEGF agents following one to three consecutive monthly loading injections (bevacizumab, ranibizumab, and aflibercept) for 6 months from January 2012 to June 2019. *Results*: We enrolled a total of 72 eyes for loading injections and the mean CRT improved from 434.04 ± 139.4 μm (before treatment) to 362.9 ± 125.0 μm after the loading injection. One week after injection, 36 eyes (50%) obtained a CRT of ≤350 μm. Fourteen eyes (19.4%) remained with a CRT of ≤350 μm for 6 months without additional injections. A total of 22 eyes (30.6%) had a CRT of >350 μm at 6 months. Fifteen eyes did not receive additional injections because of visual improvement. *Conclusions*: About 20% of DME patients can be maintained at a CRT of ≤350 μm for 6 months with only a loading injection. However, there is a tendency to delay additional injections for patients with recurrences using PRN protocol.

## 1. Introduction

Diabetic macular edema (DME) is a common cause of vision reduction in diabetic patients [1]. Because vascular endothelial growth factor (VEGF) is common in eyes with DME, various anti-VEGF agents, such as bevacizumab (Avastin, Genentech), ranibizumab (Lucentis, Genentech), and aflibercept (Eylea, Regeneron Pharmaceuticals), have been commercialized for its treatment. After a series of randomized clinical trials, they became the first-line therapy for DME treatment [2,3,4]. Because the standard treatment protocol requires frequent visits and injections, this protocol is difficult to be performed in a real-world clinical setting; and flexible strategies have been developed. In many randomized clinical trials (RCT), various injection patterns were used; including monthly injections and maintenance injections following the loading injection. Though the number of loading injections differed for each study, most of the RCT protocols stated multiple loading injections [5,6,7,8,9]. The results from these RCT studies showed the effectiveness of loading injections. As a maintenance phase, the pro re nata (PRN) protocol enables patients to receive fewer injections; in addition, an optical coherence tomography (OCT) assessment determines the timing of the injection for recurrence [7]. The treat-and-extend (TAE) protocol, which can extend the treatment interval according to therapeutic effects, is also frequently used. Many physicians believe that TAE is ideal; however, PRN is often selected in real-world clinical practice [10]. In a survey of vitreoretinal specialists in Japan, 75% of physicians chose PRN for DME treatment [11].

Thus, the PRN protocol is a popular option for anti-VEGF injections after the loading injection. However, PRN’s established scheduling protocol is not clearly defined. This may result in inappropriate timings for additional injections during the maintenance phase. In this study, we evaluate DME recurrence after the loading injections of anti-VEGF agents by the PRN protocol; we do so by using central retinal thickness (CRT) as a re-injection criterion.

## 2. Patients and Methods

This is a retrospective, observational single-center study. Our study protocol conformed to the tenets of the Declaration of Helsinki for research involving human subjects. The Institutional Review Board of Mie University Graduate School of Medicine (No. 2628) approved the protocol, which conformed to the principles of good clinical practice and the Helsinki guidelines. We registered this prospective study at http://www.umin.ac.jp (accessed on 15 September 2022, UMIN ID 000012094).

We explained the off-label use of bevacizumab to all the patients who received the bevacizumab injection. We obtained signed informed consent from all the patients. Because no other anti-VEGF agents, including ranibizumab and aflibercept, were approved before 2014 in Japan, bevacizumab was the only agent that we could use. We added other anti-VEGF agents (ranibizumab and aflibercept), after their approval in 2014, to the treatment regimens. The consent form also included a statement confirming the medical findings could be used for future research.

We examined the patients who received loading injections of anti-VEGF (bevacizumab, ranibizumab, and aflibercept) for DME between Jan 2012 and June 2019 for 6 months in the Department of Ophthalmology at the Mie University Hospital. According to the operator and patient, the number of loading injections was decided from 1 to 3.

Each patient received a comprehensive ophthalmological examination, including measurements of the best-corrected visual acuity (BCVA) and intraocular pressures; examinations of the anterior segment and fundus by slit-lamp biomicroscopy and indirect ophthalmoscopy, respectively; and macular evaluations by a Heidelberg Spectralis OCT instrument (Heidelberg Engineering Inc, Heidelberg, Germany).

### 2.1. Inclusion and Exclusion Criteria

Our inclusion criteria were: types 1 and 2 diabetes with the presence of DME (CRT greater than 350 μm in the spectral-domain OCT images), age of at least 20 years, and BCVA at a baseline of 20/320 or more. Our exclusion criteria were: prior ocular surgery within 6 months of the study, macular laser photocoagulation, and intravitreal or sub-tenon injections of steroids within 3 months before the treatment. In addition, we excluded eyes with ocular inflammation, drusen, severe proliferative diabetic retinopathy, or retinal hemorrhage; which involved the intra- or sub-foveal spaces, an epiretinal membrane, or any history of pars plana vitrectomy, glaucoma, and media opacities that considerably affected the BCVA. We also excluded patients with uncontrolled systemic medical conditions or a history of thromboembolic events.

### 2.2. Intravitreal Anti-VEGF Agent Injection

We injected the anti-VEGF agent under topical anesthesia. Each patient received 0.05 mL of anti-VEGF agents (1.25 mg of bevacizumab, 0.5 mg of ranibizumab, and 2 mg of aflibercept) intravitreally from 4 mm posterior to the corneal limbus under sterile conditions with a 30-gauge needle. After the injection, all the patients received topical levofloxacin hydrate (1.5% Cravit ophthalmic solution) for 3 days. We defined a CRT of >350 μm as a recurrence. We performed the consecutive monthly injections (loading injections) for each case. A physician decided the number of loading injections (from 1 to 3).

### 2.3. Measurement of BCVA and OCT

At every visit, we measured the BCVA with a Landolt chart. We converted the decimal BCVA to the logarithm of the minimum angle of resolution (logMAR) units for the statistical analyses. We determined the degree of DME from the images recorded by a Heidelberg Spectralis OCT instrument. Using the bundled software, we defined the CRT as the thickness between the internal limiting membrane and the retinal pigment epithelium at the central 1-mm-diameter circle of the ETDRS thickness map.

### 2.4. Statistical Analyses

We performed statistical evaluations using the Statcel 4 Statistical Program (Statcel; OMC, Saitama, Japan). We presented the results as the means ± standard deviations (SD). We used ANOVA two-way repeated measures and post hoc *t* tests with Bonferroni corrections to determine the significance of the changes in the BCVA and CRT. We considered two-tailed *p* values of <0.05 as significant.

## 3. Results

### 3.1. Baseline Characteristics

Our study consisted of 72 eyes with a baseline CRT of 350 μm or greater that received loading injections of anti-VEGF agents from 2012 to 2019. The demographics of all 72 patients were shown in Table 1. The anti-VEGF agents were: bevacizumab, ranibizumab, and aflibercept in 15, 39, and 18 eyes, respectively. A total of 20, 8, and 44 eyes received 1, 2, and 3 loading injections, respectively. Among them, 36 eyes obtained a CRT of less than 350 μm 1 week after the loading injection. A total of 14 (19.4%) and 22 (30.6%) eyes displayed a CRT of ≤350 μm (maintained group) and >350 μm (recurrence group), respectively, 6 months after the loading injection (Figure 1).

### 3.2. The Course of the Eyes after Loading Injection

Next, we present the results of our evaluation, which consisted of a detailed follow-up on 22 eyes during the first 6 months after the loading injections. Among them, 31.8% had a CRT of >350 μm at 1 month; 72.7% at 3 months; and all the patients had a CRT of >350 μm at 6 months after the loading injection (Figure 2). By 6 months, we gave an immediate additional PRN injection to 12 of the 22 eyes; and we followed up on 10 eyes without an additional injection (PRN [-]). A worsening of subjective symptoms was the main reason for the additional PRN injection.

### 3.3. BCVA and CRT Changes after Loading Injection

We present the results of the maintained group (CRT ≤ 350 μm during the 6 months) compared with the recurrence group (CRT > 350 μm) (Figure 3). We observed no significant difference for the baseline BCVA and CRT between the groups. Three monthly loading injections were performed for all the eyes in the maintained group; furthermore, there was no significant worsening of BCVA or CRT during our observations. Despite performing additional PRN injections, we observed a significant worsening of both BCVA and CRT in the recurrence group (*p* < 0.05, repeated ANOVA).

Finally, we present the results of the maintained and PRN [-] groups without additional injections (Figure 4). We observed no significant differences for the baseline BCVA and CRT between the groups. Although we observed no significant worsening of the BCVA in both groups, the CRT significantly increased for the PRN [-] group six months after the loading injections (*p* < 0.05, repeated ANOVA).

## 4. Discussion

Here, we evaluated the 6-month follow-up after the loading injection of an anti-VEGF agent for DME. A total of 36 of 72 eyes had a CRT of ≤350 μm after the loading injection, 14 eyes (19.4%) maintained a CRT of ≤350 μm, and 22 eyes (30.6%) showed a CRT of >350 μm at 6 months. Of the 22 eyes with a CRT of >350 μm, 12 required an immediate additional injection, while the other 10 did not receive an additional injection. Although they displayed an increased CRT, they did not have a worsening of BCVA; which may be the reason they required no additional injections.

In our study, 30.6% had a CRT of >350 μm; and recurrence occurred within 6 months. However, not all the patients received immediate additional injections at recurrence; this is because CRT recurrence does not always coincide with a worsening of BCVA. In general, DME patients with better BCVA tend to have thinner CRT; while those with poor vision tend to have thicker CRT. However, BCVA does not always correlate with retinal thickness [12]. There are cases with better BCVA, despite a thick CRT; and cases with poor BCVA, despite a thin CRT. The condition of the outer retinal layer (ORL), such as ellipsoid zone damage, affects BCVA more than the thickness of the retina itself. The ORL may be damaged even after complete resolution of DME. The patients with a damaged ORL and worse vision were more likely to have subsequent impaired vision after resolution of DME [13]. In real-world practice, re-injection depends on changes in the patient’s subjective symptoms. Therefore, an immediate additional PRN injection based on CRT rather than BCVA is uncommon because of patient preference. Though a survey of retinal experts recommended strict PRN protocols, they acknowledged the difficulty in implementing PRN protocol [14]. In addition, our study is based on short-term 6-month results. The long-term outcomes are unclear for patients who did not receive additional doses. Future studies on appropriate PRN protocols need to consider this factor.

In our study, 20 of 72 eyes received the initial single injection. James et al. reported 180 DME eyes treated with one or more PRN ranibizumab injections and found that approximately 1/3 of the patients only had the initial injection [15]. High-baseline BCVA contributed to the reduction in injection numbers. In our study, there was no difference in baseline BCVA between the maintained and recurrence groups. Therefore, factors other than baseline BCVA may contribute increased numbers of injections. But there is a possibility that this lack of significant difference may be by the low power of the study. We need much consideration with large numbers.

The results of the LUMINOUS study reported the analysis of real-world clinical data from approximately 4700 patients who received ranibizumab for DME. More frequent injections were associated with vision improvement, especially with loading injections [16]. The results of the MERCURY study, which analyzed the actual clinical data of ranibizumab for DME treatment in Japan, also showed the importance of the loading injection in improving BCVA [17]. In our study, almost 20% of the patients could maintain a CRT of ≤350 μm for 6 months after the loading injection alone; which suggests that the loading injection contributed to the improvement in BCVA.

However, our study has several problems. First, our study had a small sample size; and we did not perform an analysis comparing the three anti-VEGF agents. Second, although we defined recurrence as a retinal thickness of 350 μm or greater, depending on the morphology, there are some cases in actual clinical practice where good BCVA is maintained with retinal thicknesses greater than 350 μm. A more detailed examination of the retinal structure, i.e., status of the outer retinal layer, is needed. Third, we used CRT as a re-injection criterion. However, some studies have reported that peripheral macular thickness can reflect visual function more sensitively compared with the central macular region [18,19]. We also did not evaluate the DME sub-type (i.e., subretinal fluid or intraretinal fluid pattern) for each case. Because we did not evaluate retinal thickness outside of the central area or DME sub-type, it is not clear whether these findings relate with the re-injection pattern; we need further evaluation about this matter. Finally, the follow-up period of 6 months is short; thus, a longer follow-up time is required.

## 5. Conclusions

In conclusion, we maintained about 20% of DME patients at a CRT of ≤350 μm for 6 months with only a loading injection of anti-VEGF agents. However, some DME patients showed a discrepancy between CRT and BCVA; this is related to the timings of additional injections. Therefore, establishing a genuine PRN protocol is important to resolve these problems. 

## Figures and Tables

**Figure 1 medicina-58-01318-f001:**
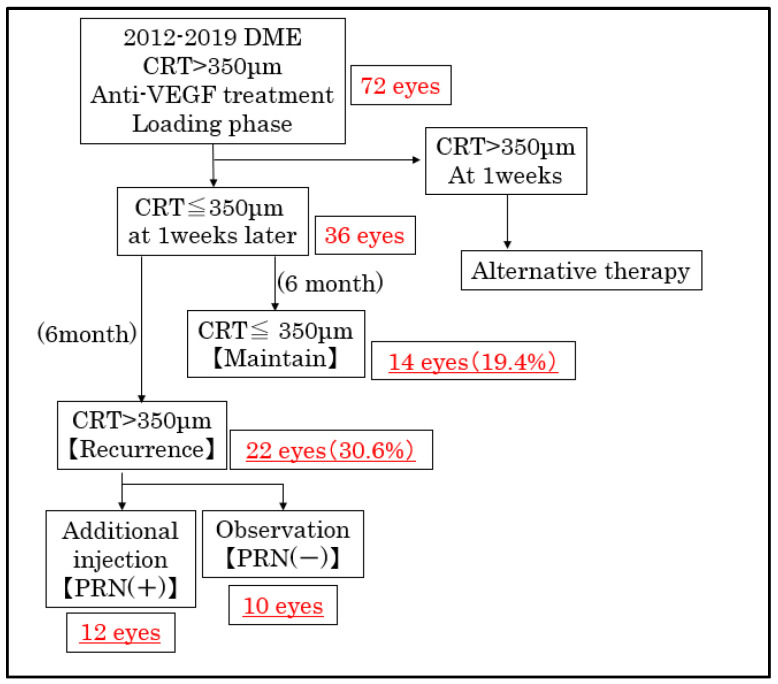
Flow chart describing the selection of the study population. CRT: central retina thickness; DME: diabetic macular edema; PRN: pro re nata.

**Figure 2 medicina-58-01318-f002:**
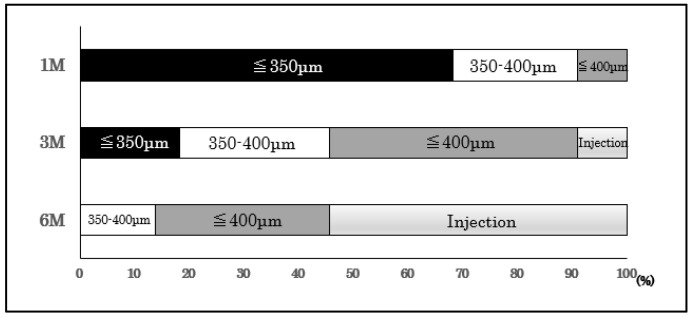
Eyes after the loading injection.

**Figure 3 medicina-58-01318-f003:**
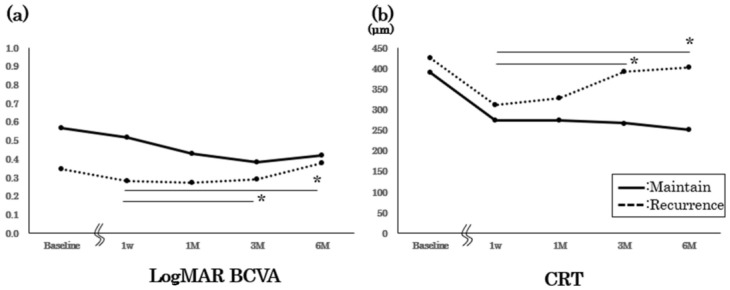
BCVA and CRT changes after the loading injections in the maintained and recurrence groups. Six-month changes after the loading injections for Log MAR BCVA (**a**) and CRT (**b**). BCVA: best-corrected visual acuity; CRT: central retina thickness. *: *p* < 0.05, repeated ANOVA.

**Figure 4 medicina-58-01318-f004:**
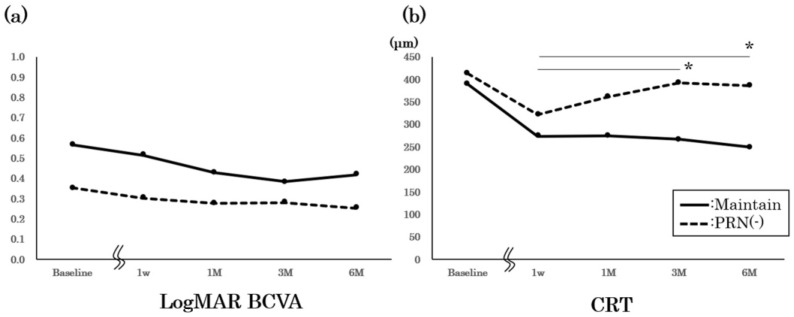
BCVA and CRT changes after the loading injections in the maintained and PRN [-] groups. Six-month changes after the loading injections for Log MAR BCVA (**a**) and CRT (**b**). BCVA: best-corrected visual acuity; CRT: central retina thickness; PRN: pro re nata. *: *p* < 0.05, repeated ANOVA.

**Table 1 medicina-58-01318-t001:** The demographics of all 72 patients.

Age (Years)	BCVA (log MAR)		CRT (μm)	
	Baseline	After Loading Phase	Baseline	After Loading Phase
64.4 ± 9.7	0.54 ± 0.30	0.46 ± 0.25	434.0 ± 139.4	362.9 ± 125.0

BCVA, best corrected-visual acuity; CMT, central macular thickness; log MAR, logarithm of the minimum angle of resolution. After the loading phase; 1 week after the loading injection.

## Data Availability

The datasets used during the current study are available from the corresponding author on request.

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
