# Peer review of "Progress of Diabetic Macular Edema after Loading Injection of Anti-Vascular Endothelial Growth Factor Agents in Real-World Cases"

_medicina, 2022, doi:10.3390/medicina58101318_

Round 1

Reviewer 1 Report

This study is very simple. Few cases, long accrual period, easy statistics, clear conclusion.  I would like to add just few data on patients, only eyes are considered in the paper.

Due to the low number of observation, authors should include in the 'Discussion' (lines 174-175) that the lack of significant differences could be caused by the low power of the study.

Author Response

Reply to reviewer report 1

Q1) This study is very simple. Few cases, long accrual period, easy statistics, clear conclusion.  I would like to add just few data on patients, only eyes are considered in the paper.

A) We add patient back ground data of the patients as Table 1.

Q2) Due to the low number of observation, authors should include in the 'Discussion' (lines 174-175) that the lack of significant differences could be caused by the low power of the study.

A) We add about this in L195-.

Reviewer 2 Report

Overview: The authors have conducted a study on the benefits of the loading dose of anti-VEGFs on DME. The loading dose was administered on PRN basis for 6 months. Of 72 eyes enrolled for loading injections, 50% obtained a CRT of ≤350μm at 1 week. 19.4% remained with a CRT of ≤350μm for 6 months without additional injections, and 30.6% had a CRT of >350μm at 6 months. Fifteen eyes did not receive additional injections because of visual improvement. They concluded that about 20% of DME patients can be maintained at a CRT of ≤350μm for 6 months with only a loading injection.

This is well-written nice article, which will be helpful clinically. If we can extend the anti-VEGF injections, it will help to de-congest the busy clinics and help improve the eye care services. It needs some corrections as mentioned in the specific comments below.

Specific comments:

Abstract: It is not very clear about the loading dose. Was it done for 6 months in all cases on PRN basis?

Q: What was the mean CRT prior to the start of the loading dose and what was the mean CRT reduction?

Introduction:

Q: Kindly clarify the temporal definition of loading dose of anti-VEGF? For example, in your study you have mentioned six months in L15.

Methods:

L55: Prior to 2014, only bevacizumab was used in Japan. Here you need to clarify that post 2014 other anti-VEGF such as ranibizumab and aflibercept were added to the treatment regimens because your study period was 2012 to 2019.

L59: As mentioned above the definition of loading dose is not well defined. In Abstract L15 you have mentioned 6 months, and in Methods L59 you have mentioned more than 6 months. Or should it be defined based on the number of initial injections as in L61-62 you have mentioned from 1 to 3 injections. Please address this differences.

L66: Please provide details of the OCT machine used.

Q: Have you included both the types 1 and 2 diabetes?

Q: You have taken CRT as the main re-injection criterion. But you may agree that off-centre DME are very common and these also affect vision. What is your logic for taking only CRT as re-injection criterion?

In fact, some studies have reported that peripheral macular region thickness is more correlated with the vision than that of the central macular region. You may refer:

https://doi.org/10.1167/tvst.10.13.32

https://doi.org/10.1167/tvst.10.2.10

L79: Please specify the term local anaesthesia. Is it per- or retrobulbar or topical or sub-tenon? Anti-VEGF injection is usually performed under topical anaesthesia.

L79: In addition to mentioning 0.05 ml of anti-VEGF, please also mention the amount of each anti-VEGF injected. Anti-VEGF dosing for DME and other diseases may be different.

L89-90: This OCT details should have been mentioned in the first mention in L66.

Results:

Demographic findings are missing in the Results. Please mention them.

L117: worsening subjective symptoms was the main reason for additional PRN injection. How much weightage did you give to the OCT findings such as CRT or off-centre retinal thickness or fluid collection?

Discussion:

L148-154: Here I think you have summarized the main findings. It looked more like a Result section, but I think it is OK.

L160-163: Please provide references here. Here you may also add a paragraph on structural and functional correlation in DME.

L170: It should be James et. al., not James alone.

Thank you. I enjoyed going through your paper.

Author Response

Reply to reviewer report 2

Q: Abstract: It is not very clear about the loading dose. Was it done for 6 months in all cases on PRN basis?

A: We add description about this at L14 and 15.

Q: What was the mean CRT prior to the start of the loading dose and what was the mean CRT reduction?

A: We add description about this at L16.

Introduction:

Q: Kindly clarify the temporal definition of loading dose of anti-VEGF? For example, in your study you have mentioned six months in L15.

A: We add description about loading phase at L35-.

Methods:

Q: L55: Prior to 2014, only bevacizumab was used in Japan. Here you need to clarify that post 2014 other anti-VEGF such as ranibizumab and aflibercept were added to the treatment regimens because your study period was 2012 to 2019.

A: We add description about this at L63-.

Q: L59: As mentioned above the definition of loading dose is not well defined. In Abstract L15 you have mentioned 6 months, and in Methods L59 you have mentioned more than 6 months. Or should it be defined based on the number of initial injections as in L61-62 you have mentioned from 1 to 3 injections. Please address this differences.

A: We delete the words ”more than” from this sentence (L68).

Q: L66: Please provide details of the OCT machine used.

A: We add details of OCT machine at L74.

Q: Have you included both the types 1 and 2 diabetes?

A: Both type I and II patients were included (L77).

Q: You have taken CRT as the main re-injection criterion. But you may agree that off-centre DME are very common and these also affect vision. What is your logic for taking only CRT as re-injection criterion?

In fact, some studies have reported that peripheral macular region thickness is more correlated with the vision than that of the central macular region. You may refer:

https://doi.org/10.1167/tvst.10.13.32

https://doi.org/10.1167/tvst.10.2.10

A: We add this matter as limitation and also add these two paper as references (L211-).

Q: L79: Please specify the term local anaesthesia. Is it per- or retrobulbar or topical or sub-tenon? Anti-VEGF injection is usually performed under topical anaesthesia.

A: We delete “topical” and exchange “topical” (L88).

Q: L79: In addition to mentioning 0.05 ml of anti-VEGF, please also mention the amount of each anti-VEGF injected. Anti-VEGF dosing for DME and other diseases may be different.

A: We add description of the amount of each agents (L89).

Q: L89-90: This OCT details should have been mentioned in the first mention in L74.

Q: We move this to L74.

Results:

Q: Demographic findings are missing in the Results. Please mention them.

A: We add patient’s demographics in Table 1.

Q: L117: worsening subjective symptoms was the main reason for additional PRN injection. How much weightage did you give to the OCT findings such as CRT or off-centre retinal thickness or fluid collection?

A: We evaluate only CRT, not off-centre retinal thickness or fluid collection such as SRF and IRF. We add description about this as limitation (L213-).

Discussion:

Q: L148-154: Here I think you have summarized the main findings. It looked more like a Result section, but I think it is OK.

A: Thanks for your suggestion.

Q: L160-163: Please provide references here. Here you may also add a paragraph on structural and functional correlation in DME.

A: We add reference and description about this at L)180.

Q: L170: It should be James et. al., not James alone.

A: We correct this (L190).
